# Quantifying absolute addressability in DNA origami with molecular resolution

Maximilian T. Strauss[1,2], Florian Schueder[1,2], Daniel Haas[1,2], Philipp C. Nickels [1,2] & Ralf Jungmann[1,2]

Self-assembled DNA nanostructures feature an unprecedented addressability with sub-nanometer precision and accuracy. This addressability relies on the ability to attach functional entities to single DNA strands in these structures. The efficiency of this attachment depends on two factors: incorporation of the strand of interest and accessibility of this strand for downstream modification. Here we use DNA-PAINT super-resolution microscopy to quantify both incorporation and accessibility of all individual strands in DNA origami with molecular resolution. We find that strand incorporation strongly correlates with the position in the structure, ranging from a minimum of 48% on the edges to a maximum of 95% in the center. Our method offers a direct feedback for the rational refinement of the design and assembly process of DNA nanostructures and provides a long sought-after quantitative explanation for efficiencies of DNA-based nanomachines.

---

[1] Department of Physics and Center for Nanoscience, Ludwig Maximilian University, 80539 Munich, Germany. [2] Max Planck Institute of Biochemistry, 82152 Martinsried near Munich, Germany. These authors contributed equally: Maximilian T. Strauss, Florian Schueder, Daniel Haas. Correspondence and requests for materials should be addressed to R.J. (email: jungmann@biochem.mpg.de)

Structural DNA nanotechnology[1,2] has revolutionized the field of molecular self-assembly by harnessing the programmability and specificity of DNA hybridization for sequence-guided self-assembly. DNA origami[3], in particular, marked a breakthrough, allowing researchers to readily design and build structures of almost arbitrary shape and complexity[4–11]. In DNA origami, a long single strand (the "scaffold") is folded into a pre-designed shape by ~200 short, complementary strands (the "staples"). Each staple has a unique sequence and specifically binds parts of the scaffold together during thermal annealing, thus folding the scaffold into the pre-designed shape. While the large variety of shapes constructed to-date is impressive, the true power of DNA origami lies in the addressability of specific sites on the structure with sub-nanometer precision and accuracy[12–16] via the modification of single staples. Successful addressability of a staple is directly linked to its incorporation and accessibility: the staple has to be incorporated efficiently and it needs to be accessible for downstream attachment of guest molecules, e.g., via complementary strand hybridization or direct chemical modification. Hence, it is necessary to characterize both factors on the single-staple level with absolute quantification. Although, recent studies assessed the overall structural integrity using bulk gel assays[17] and the relative abundance of single staples using next-generation sequencing[18], we still lack the ability to quantify incorporation and accessibility in an absolute manner on the level of single staples. Recent advancements in optical super-resolution microscopy[19] allow for precise, noninvasive characterization of objects below the diffraction limit of light. Specifically, DNA Points Accumulation in Nanoscale Topography (DNA-PAINT)[20,21] super-resolution microscopy is well-suited to characterize DNA nanostructures because it can achieve the thus far unprecedented spatial resolution of ~5 nm, enabling the quantification of the accessibility and absolute incorporation efficiency of every single staple in a DNA origami structure[22].

## Results

**In silico validation of the method**. Transient, repetitive binding of dye-labeled oligonucleotides ("imager" strands) to their complementary targets ("docking" sites) can be observed as apparent blinking (Fig. 1a). The apparent blinking is used to reconstruct super-resolution images that visualize the designed pattern of docking sites, e.g., a 20-nm-grid structure (based on the two-dimensional (2D) rectangular DNA origami, details about the design are shown in Supplementary Figs. 1 and 2). These 20-nm-grids, however, stochastically miss reconstructed points at designed sites[21,23] because the docking sites at these missing points are not transiently visited by an imager strand. This could be explained by two mechanisms: (1) staples are not incorporated into the structure; or (2) staples are incorporated, but docking sites are not available for binding of imager strands (i.e., sequestered). As a result, these positions are not accessible for downstream modification (Fig. 1b). To assess this accessibility, we developed a software tool that detected DNA origami structures in a reconstructed DNA-PAINT image and subsequently aligned them to a template structure to create a sum image (Fig. 1c and Supplementary Figs. 3 and 4). Then, we quantified the number of localizations at each docking site in the sum image. As the binding of imager strands to their docking sites is repetitive, we defined a minimum number of localizations as a threshold value: sites with values below this threshold were classified as not detected (Fig. 1c). To validate our analysis workflow, we performed in silico DNA-PAINT experiments of 20-nm-grids with an occurrence probability of individual docking sites ranging from 30 to 100% using the software program Picasso[21]. The number of detected docking sites was in good agreement with the

number of simulated docking sites (Fig. 1d), which confirmed the applicability of our analysis approach to measure the detectability of single docking sites.

**Investigating incorporation and accessibility**. Next, we decoupled the two possible underlying mechanisms for non-detectable docking sites: (1) incorporation and (2) accessibility. In order to assay each mechanism, we designed a 20-nm-grid carrying staples that are simultaneously extended with orthogonal docking sites on the 3′- and 5′-end (Fig. 2a). We then performed sequential two-color DNA-PAINT imaging and interactively evaluated 100 origami structures with a total of 1200 designed docking sites (the imaging results are shown in Fig. 2b and Supplementary Fig. 5). In 78.5% of the cases, we detected a signal from both docking sites (3′- and 5′-end). In 5%, we only detected the 3′-end site, and in 7.2%, only the 5′-end site. No site at all was detected in 9.3% of all cases. Since the detection of 3′- and 5′-ends should be independent of each other, we estimated that there was a ~2.4% probability that neither the 3′-end nor the 5′-end of an actually incorporated staple was detected. Therefore, we concluded that in the 9.3% of the cases (in which no site was detected) the staple was indeed not incorporated. Ultimately, this allowed us to assess the accessibility of docking sites for downstream studies, as well as to quantify the actual incorporation efficiency of single staples. To translate accessibility to absolute staple incorporation efficiency, we added an offset of +7% when imaging the accessibility of a 3′-end site (since we typically use 3′-end extensions as docking sites in DNA-PAINT, we concentrated on 3′-ends for the following experiments). Nevertheless, as the vision of structural DNA nanotechnology[24] is to arrange matter in a prescribed manner by site-specific attachment of molecular entities, we believe the ultimate measure of quality for DNA origami should be the accessibility of docking sites. Therefore, we focus on detection efficiencies (i.e., accessibility) for the rest of this study.

**Assembly conditions and staple detection**. We investigated the influence of different assembly conditions on the detection efficiency of individual staples. First, we evaluated the number of detected sites as a function of the annealing time: for the 20-nm-grid, we tested annealing times ranging from 5 min to 3 days (Fig. 3b) and determined detection efficiencies for all 12 staple positions separately. Since the 20-nm-grid has rotational and mirror symmetry, we then averaged over all 12 sites of a single grid to obtain an average detection efficiency per origami. We measured an average detection efficiency of ~82% for all annealing times (standard deviation $\sigma = 1.3\%$), underlining the remarkable robustness of the 2D rectangular DNA origami[25]. Additionally, magnesium concentrations between 8 and 16 mM during folding, as well as the long-term storage at room temperature of purified 20-nm-grids did not result in any significant change in detection efficiency (Supplementary Figs. 6 and 7). Second, we investigated the detection efficiency of staples with respect to their molar excess over the scaffold ranging from ten times to ~500 times molar excess (Fig. 3c and Supplementary Fig. 8). The efficiency improved by more than 10% when increasing the excess from ten times to ~500 times (from 72 to 84% on average) and followed a Michaelis–Menten kinetic (for fit parameters see Supplementary Table 1).

**Quantifying the accessibility of individual docking sites**. Next, we moved away from an average measure of detection efficiency and tested our capability to accurately quantify the accessibility of single docking sites. Accordingly, we designed two distinct structures that break the rotational and mirror symmetry of the regular 20-nm-grid. Each structure carried an alignment pattern

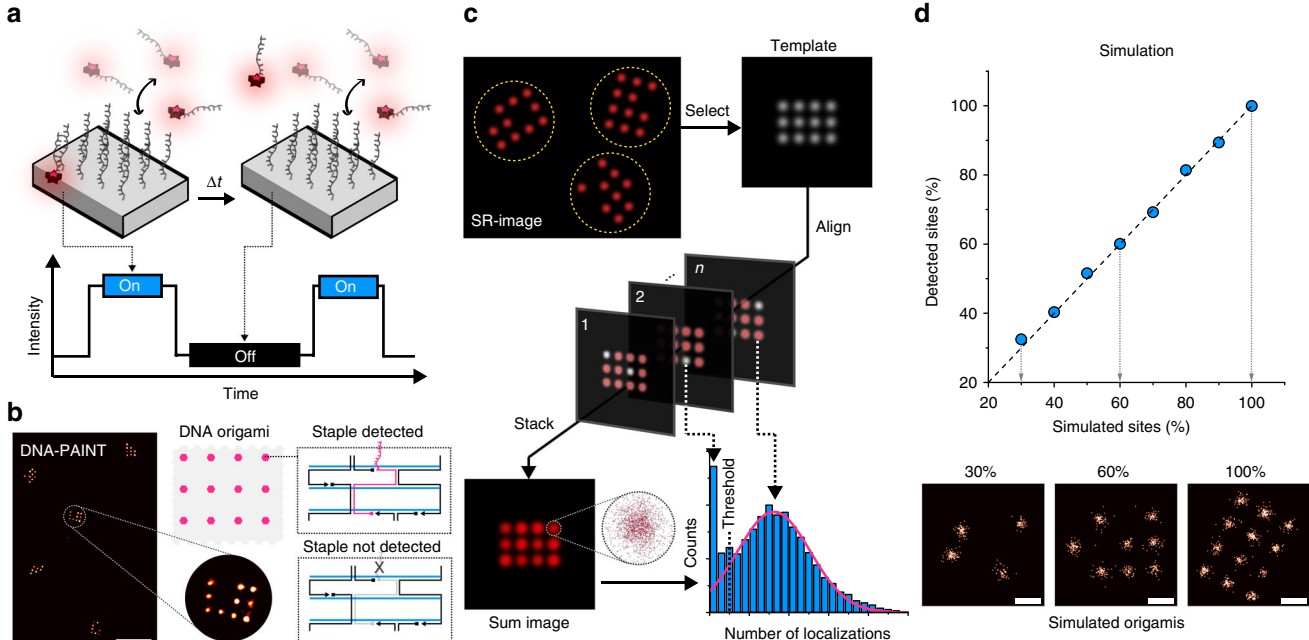

**Fig. 1** Quantification of detected staples in DNA origami using super-resolution. **a** DNA-PAINT concept: transient hybridization of dye-labeled imager strands to docking sites on DNA origami enables super-resolution imaging. **b** Typical DNA-PAINT image of 20-nm-grids allows distinction of individual binding sites. Zoom-in of a 20-nm-grid shows that some of the 12 grid sites were not detected. **c** Quantification workflow for assessing abundance of docking sites. Single structures were selected and aligned to a template of the designed grid structure by image cross-correlation. Subsequently, a histogram of localizations per grid site was used to determine a cut-off threshold below, which a site is defined as not detected. This threshold is defined as the number of localizations at half-maximum as determined by the Gaussian fit (magenta curve). **d** Benchmarking of the detection workflow was performed using simulated ground-truth DNA-PAINT data. Blue dots represent simulated 20-nm-grids with varying percentage of simulated docking sites and their evaluation result. The dashed line is the identity line. Scale bars: 100 nm in **b**, 20 nm in **d**

in the form of an arrowhead and a line, as well as three (3 binding sites (BS)) or six (6 BS) docking sites (Fig. 3d). These two structures enabled us to probe the following: first, false positives (i.e., detected sites in 3 BS that are only present in 6 BS); second, the ability to site-specifically probe the detection efficiency. We first tested for false positives and detection efficiencies of single sites using in silico DNA-PAINT data and detected 0% false positives and 100% of the simulated true positive staples (number of simulated structures $n = 50$). Then, we performed in vitro DNA-PAINT experiments (Supplementary Figs. 9 and 10) and detected an average of 2% false positives (number of analyzed structures $n = 250$). Furthermore, the standard deviation between detection efficiencies of the same sites on the two different structures was ~2.6%. These findings suggested that there are no systematic errors induced by our approach, emphasizing our ability to quantify absolute numbers of detected sites in DNA origami structures. At the same time, we measured a large difference (>10%) in the percentage of detected sites for positions 4–6 on both 3 BS and 6 BS structures (in the range of 72–83%), which indicated that there is a positional dependency of detection and therefore staple incorporation.

**Quantifying every staple in a DNA origami structure.** Finally, this led us to quantify the accessibility of every single staple in the 2D rectangular DNA origami structure. We designed a total of 18 different rectangles, each comprising of an alignment pattern (arrowhead + line) and a unique arrangement of 12 detection sites per rectangle (Fig. 4a and Supplementary Fig. 11) to allow template identification for each structure (Fig. 4b). These 18 unique designs enabled us to individually probe a total of 168 staples in a single DNA-PAINT acquisition experiment (Supplementary Figs. 12 and 13). We deliberately left out staple

strands surrounding the biotinylated strands for surface attachment (white hexagons in Fig. 4c), as staple orientation and routing at these locations are inconsistent with the standard design of the rectangle. We then quantified the accessibility for all 168 staples and constructed a heatmap of the rectangle displaying the accessibility as a function of docking-site position (Fig. 4c). The results indicate a consistently lower efficiency of detection on the outside of the structure (with a minimum of 41%) compared to inner areas where detection efficiencies reached 88% (the average detection efficiency for all strands was 77%). Taking the detection efficiency offset of 7% determined by the results from Fig. 2, this translates to absolute incorporation efficiencies of 48–95% with an average of 84%, in good agreement with qualitative results of relative staple abundance from next-generation sequencing[26]. A heatmap displaying the translated values of absolute incorporation efficiencies is shown in Supplementary Fig. 14.

To further evaluate our assumption that we can directly translate the detection efficiency offset from Fig. 2 to strand incorporation and eliminate possible accessibility effects, we targeted a single-stranded section of the scaffold strand (Supplementary Fig. 16a) on the edge of the 2D rectangular origami (scaffold loop). The assumption was that this scaffold loop must be present in every structure. We detected the scaffold loop in 90% of all cases. To further investigate the effect of potential surface interaction, we used a three-dimensional DNA origami structure, the force clamp[27]. This structure spans a section of the scaffold strand between two pillars more than 20 nm above the surface. Here, we mimicked the single-stranded scaffold loop we targeted in the 2D rectangular origami (Supplementary Fig. 16b) and found a 91% detection efficiency, indicating that there is no significant effect arising from surface interactions. We additionally used the force-clamp structure to

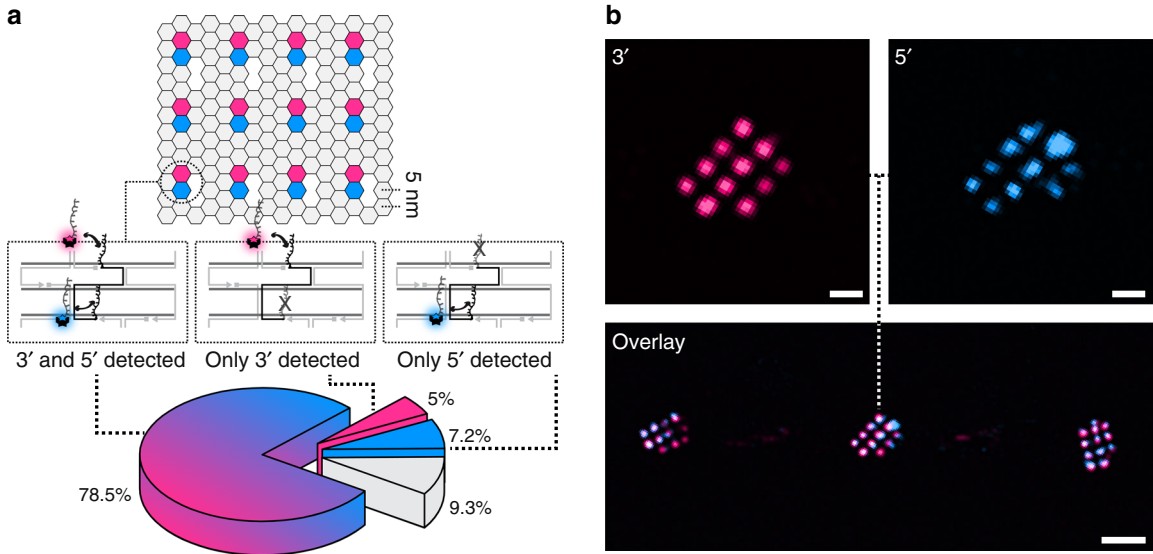

**Fig. 2** Experimental validation of accessibility and incorporation efficiency. **a** The 20-nm-grid staples were extended with orthogonal DNA-PAINT docking sites on the 3′-end (magenta) and 5′-end (blue) and subsequently imaged using Cy3B-(magenta) and Atto647N-labeled (blue) imager strands. The pie chart shows the percentage of docking sites where both ends were detected (blue to magenta slice), only the 3′-end (magenta slice), only the 5′-end (blue slice), and no end was detected at all (gray slice). **b** Overlay and color-separated zoom-ins of the two-color DNA-PAINT measurement. Scale bars: 20 nm in **b** (top), 100 nm in **b** (bottom)

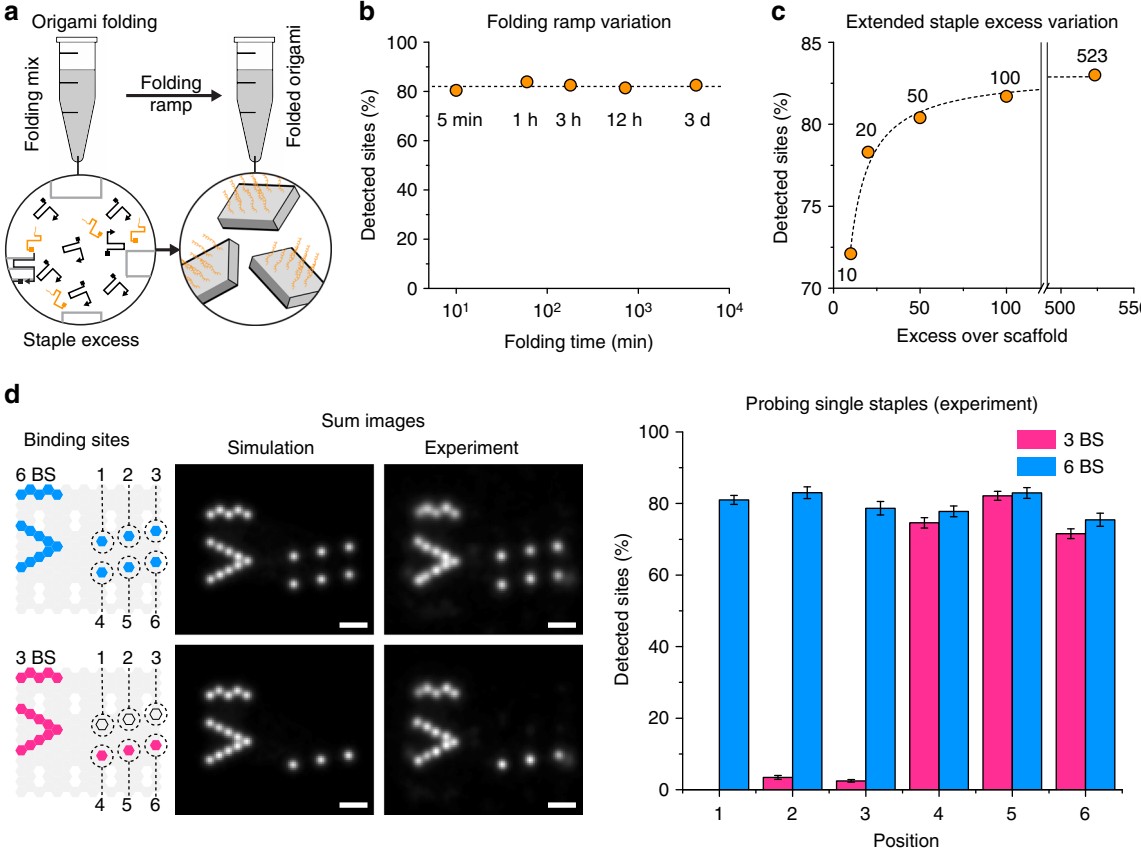

**Fig. 3** Influence of folding conditions and experimental validation of accessibility of single sites. **a** Schematic folding of 20-nm-grids with the staples that carry detection sites in orange. **b** Percentage of detected sites as a function of folding time. The dashed line represents the mean of all 5 measurements (mean: ~82%). **c** Percentage of detected sites as a function of molar staple excess over scaffold. The dashed line represents the fit of a Michaelis–Menten curve (saturation: 83%). **d** Schematics of the 6 BS (blue) and 3 BS (magenta) structures with the arrowhead + line alignment pattern, their DNA-PAINT sum images, and the percentage of detected sites depending on the position. Error bars represent the standard deviation and were generated by repeated ($n = 10$) random selection of a subset of 250 structures from all selected structures of the whole field of view. Scale bars, 20 nm in **d**

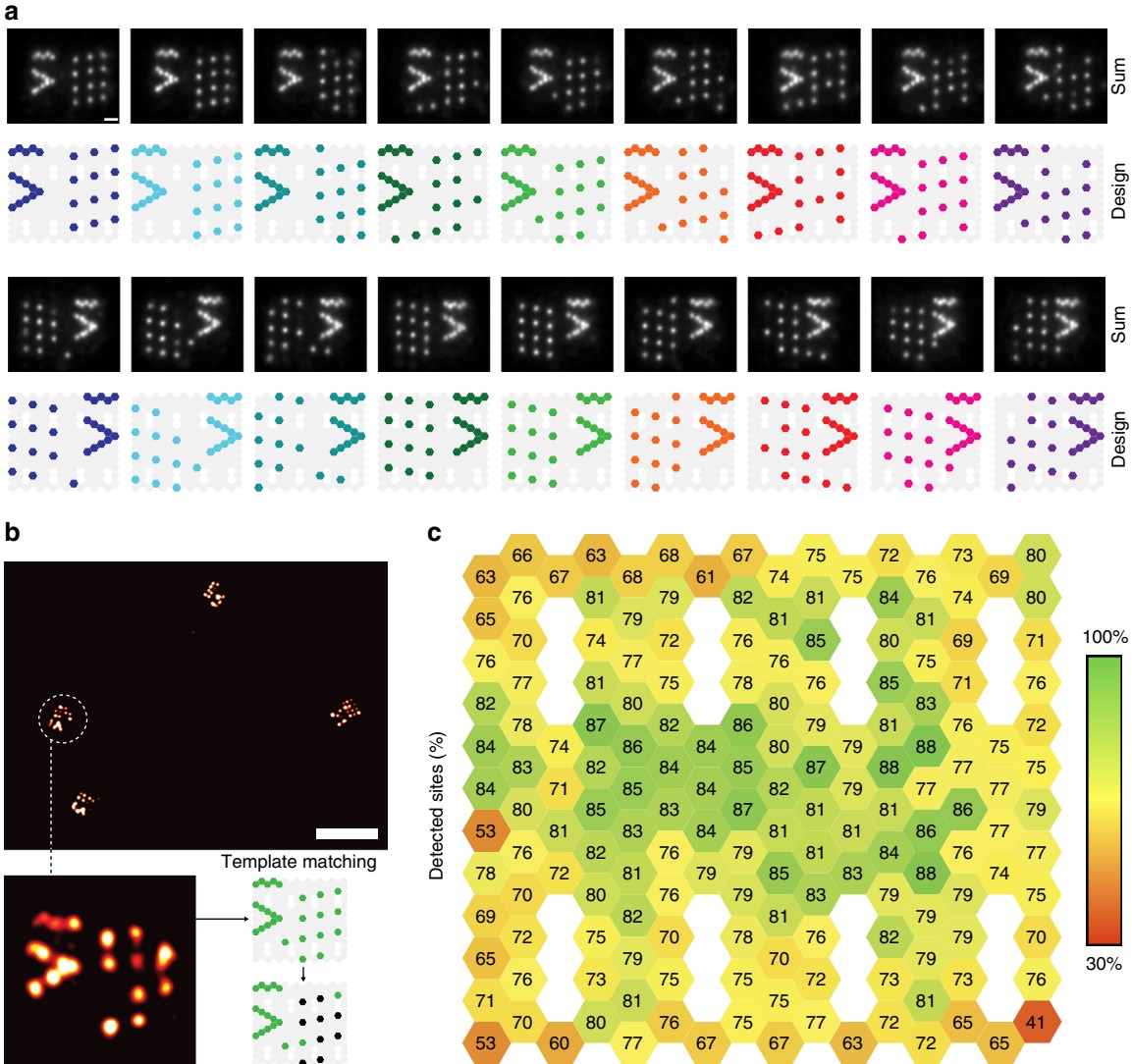

**Fig. 4** Accessibility of all staples in a 2D rectangle. **a** A total of 18 design variants were used to probe all addressable staples in a 2D rectangular DNA origami structure in a single experiment. Designed patterns and sum images of experimentally obtained DNA-PAINT images are shown. **b** Arrowhead + line alignment patterns allowed the unique assignment of a detected structure to the design template. Shown in the zoom-in is a single structure matched to a template with black sites identified as detected. **c** Heatmap of 168 individually probed staples of the 2D rectangle, generally showing higher detection efficiencies in the center of the structure and lower detection efficiencies towards the edges. Average detection efficiency: 77% (corresponding to an average incorporation efficiency of 84%). Scale bars, 20 nm in **a**, 100 nm in **b**

directly investigate the effect of accessibility on the detection efficiency. For this we modified the force clamp structure to exhibit a slightly stretched and thus ideally accessible scaffold section (Supplementary Fig. 16c). The measured detection efficiency of 97% suggests that the maximum error arising from accessibility is 3%. To translate our results back to incorporation, we hybridized a staple to the scaffold between the two pillars of the force clamp (Supplementary Fig. 16d). The 32 nt staple (same length as the staples in the 2D rectangular origami) was extended with the 3′-end docking site used throughout this study. Here, we measured a detection efficiency of the docking site of 94%, suggesting an error of 3% caused by incomplete incorporation. Ultimately, we conclude that the offset determined in Fig. 2 can be used to directly estimate incorporation from measured detection efficiency.

The transient binding of imager strands to their docking sites should not be affected by the position of the docking site on the DNA origami bound to flat surface[20,26]. Therefore, we argue that the change in detection efficiency as a function of docking site

position is indeed an effect of an underlying change in incorporation efficiency. This could be explained by the fact that staples at the edges and corners are missing neighboring helices and/or lack stacking interactions to neighboring strands. This hypothesis is further supported by qualitatively comparing our heatmap to finite-element-based modeling (Supplementary Fig. 15) of thermal fluctuation of the same rectangular origami using the software tool Cando[28]. Further quantitative assessment of these effects could be achieved by sequence-level coarse-grained[29] or fully atomistic[30] molecular dynamics simulations.

## Discussion

In recent years, many DNA origami-related studies reported on the attachment yield of various functional entities, such as streptavidin[16,20], DNA walkers[31,32], gold nanoparticles[33], motor proteins[34], and DNA strands[23,35]. We collected eight values for reported yields and translated these values into incorporation efficiencies of single staples (Supplementary Table 2). The translated incorporation values are either in good agreement with

our average incorporation efficiency of 84% or match our measured maximum incorporation of 95%. Thus, our study now provides a quantitative explanation of these reported attachment yields. From our results, we derive two recommendations for researchers planning to use DNA origami structures for the arrangement of functional entities. First, staples that are used as sites for downstream modification should be included in at least a 50 times molar excess over the scaffold to maximize the incorporation and thus efficiency of downstream modifications. Second, attachment points of downstream modifications, as well as tracks for hybridization cascades (e.g., for DNA walkers of localized chemical reaction networks) based on the rectangular origami design should be placed at points of high incorporation efficiency (Fig. 4c) and should—if possible—be designed redundantly (as already established for the attachment of metallic nanoparticles[36]).

In conclusion, we presented a method for the absolute quantification of single strand incorporation and downstream modification accessibility by using the unique capabilities of DNA-PAINT super-resolution microscopy to achieve single-staple-level resolution (we achieved a maximum localization precision of 1.37 nm as calculated by a nearest neighbor based analysis[37], see Supplementary Table 3). We believe that our method will allow for rational engineering of the design and assembly process of DNA nanostructures in order to maximize downstream attachment. This method is not limited to the 2D rectangular DNA origami structure shown here, but can be applied to virtually any DNA-nanostructure geometry. Additionally, this approach can be directly used to characterize the labeling efficiency of antibodies or cellular proteins and nucleic acids, potentially making it of great interest for super-resolution microscopy in general and quantitative structural biology in particular.

## Methods

**Materials and buffers.** Unmodified, dye-labeled, and biotinylated DNA oligonucleotides were purchased from MWG Eurofins or Integrated DNA Technologies. DNA scaffold strands were purchased from Tilibit (p7249, identical to M13mp18). Streptavidin was purchased from Thermo Fisher (catalog number: S-888). Bovine serum albumin (BSA) and BSA-biotin obtained from Sigma-Aldrich (catalog number: A8549). Glass slides and coverslips were purchased from Marienfeld (cat. no. 0107032) and Thermo Fisher (cat. No. 10756991). Freeze 'N Squeeze columns were ordered from Bio-Rad (cat. no. 732-6165). Polyethylene glycol (PEG)-8000 was purchased from Merck (cat. No. 6510-1KG). Four buffers were used for sample preparation and imaging: buffer A (10 mM Tris-HCl pH 7.5, 100 mM NaCl, 0.05% Tween 20, pH 7.5); buffer B (5 mM Tris-HCl pH 8, 10 mM MgCl₂, 1 mM EDTA, 0.05% Tween 20, pH 8); buffer O (5 mM Tris-HCl pH 8, 12.5 mM MgCl₂, 1 mM EDTA, 0.05% Tween 20, pH 8) and buffer O⁺ (same as O, but supplemented with 1× PCA, 1× PCD, and 1× Trolox). A concentration of 100× Trolox: 100 mg Trolox, 430 μl 100% Methanol, 345 μl 1 M NaOH in 3.2 ml H₂O. A concentration of 40× PCA: 154 mg PCA, 10 ml water, and NaOH were mixed and adjusted to pH 9.0. 100× PCD: 9.3 mg PCD, 13.3 ml of buffer (100 mM Tris-HCl pH 8, 50 mM KCl, 1 mM EDTA, 50% Glycerol). PEG-buffer was used for PEG precipitation[37] (15% PEG-8000, 500 mM NaCl, 12.5 mM MgCl₂ in TAE pH 8.0).

**Super-resolution microscopy setup.** Fluorescence imaging was carried out on an inverted Nikon Eclipse Ti microscope (Nikon Instruments) with the Perfect Focus System, applying an objective-type total internal reflection fluorescence (TIRF) configuration with an oil-immersion objective (Apo SR TIRF 100×, NA 1.49, Oil). Two lasers were used for excitation: 561 nm (200 mW, Coherent Sapphire) or 640 nm (150 mW, Toptica iBeam smart). The laser beam was passed through cleanup filters (ZET561/10 or ZET642/20, Chroma Technology) and coupled into the microscope objective using a beam splitter (ZT561rdc or ZT647rdc, Chroma Technology). Fluorescence light was spectrally filtered with an emission filter (ET600/50m and ET575lp or ET705/72m and ET665lp, Chroma Technology) and imaged on an electron-multiplying charge-coupled device (EMCCD) camera (Andor iXon Ultra 897, used for Figs. 1a and 3c) or sCMOS camera (Andor Zyla 4.2, used for Figs. 3b, d, and 4) without further magnification, resulting in an effective pixel size of 160 nm (EMCCD) or 130 nm (sCMOS after 2 × 2 binning).

**DNA origami self-assembly.** Self-assembly of DNA origami was accomplished in a one-pot reaction mix with 40 μl total volume, consisting of 10 nM scaffold strand, 100 nM folding staples, 10 nM biotinylated staples (500 nM for Fig. 2 and

Supplementary Fig. 16d), and 1 μM (Figs. 1b, 2, 3b, and 4) or varying concentrations (Fig. 3c) of docking site strands (5′-staple-TTATACATCTA-3′) in folding buffer (1× TE buffer with 12.5 mM MgCl₂). The reaction mix was then subjected to a thermal annealing ramp using a thermocycler (Mastercycler Nexus Gradient, Eppendorf or Tetrad 2, Bio-Rad). If not otherwise noted, the reaction mix was first incubated at 80 °C for 5 min and then cooled from 60 to 4 °C in steps of 1 °C per 3.21 min and then held at 4 °C until stored at −20 °C protected from light. Samples for the measurement with varying staple excess (Fig. 3c) were purified via three rounds of PEG precipitation by adding the same volume of PEG-buffer, centrifuging at 10,000×g at 4 °C for 30 min, removing the supernatant, and resuspending in folding buffer. Structures for Fig. 2 were gel purified by mixing with 1× loading dye and subsequently subjected to agarose gel electrophoresis (1.5% agarose, 0.5× TAE, 10 mM MgCl₂, 1× SYBR Safe) at 3 V cm⁻¹ for 3 h. Gel bands were extracted, crushed, filled into a Freeze 'N Squeeze column, and centrifuged for 5 min at 1000×g at 4 °C. Force-clamp structures were designed and assembled as described before[27]. The staple excess of the extended staple in the structure of Supplementary Fig. 16d was 100× over the scaffold.

**Sample preparation.** For sample preparation, a piece of coverslip and a glass slide were sandwiched together by two strips of double-sided tape (Scotch, cat. no. 665D) to form a flow chamber with inner volume of ~20 μl. First, 20 μl of biotin-labeled bovine albumin (1 mg/ml, dissolved in buffer A) was flushed into the chamber and incubated for 2 min. The chamber was then washed with 40 μl of buffer A. A volume of 20 μl of streptavidin (0.5 mg ml⁻¹, dissolved in buffer A) was then flushed through the chamber and allowed to bind for 2 min. After washing with 20 μl of buffer A and subsequently with 20 μl of buffer B, 20 μl of biotin-labeled DNA structures (~100–400 pM, see Supplementary Table 4) in buffer B were flushed into the chamber and incubated for 2 min. The chamber was washed with 40 μl of buffer B. Finally, 20 μl of the imager solution was flushed into the chamber, which was subsequently sealed with epoxy (Toolcraft, cat. no. TC-EPO5-24) before imaging.

**Imaging conditions.** Refer to Supplementary Table 4 for an overview of imaging conditions and Supplementary Table 5 for the imager strand sequences. Supplementary Table 3 shows the super-resolution data properties of all measurements.

**Data simulation.** In silico experiments were performed using the simulation module of the Picasso[18] software package. For Fig. 1c, 50 structures consisting of 12 BS spaced 20 nm apart (the "20-nm-grid" structure) were simulated with varying incorporation of staple strands (30–100%). Further simulation parameters were an image size of 128 × 128 px and an acquisition time of 50 min (15,000 frames at 200 ms integration time). For the simulation of the 3 BS and 6 BS structure of Fig. 3d, again 50 structures with the same parameters were simulated. For each simulation run, a dark time of 12.5 s and a bright time of 0.5 s were used, corresponding to 5 nM imager strand concentration at a constant association rate of $1.6 \times 10^6$ M⁻¹ s⁻¹. Further simulation parameters were a pixel size of 160 nm, a detection rate of 35 photons × ms⁻¹ kW⁻¹ cm⁻², a budget of $1.5 \times 10^6$ photons, a power density of 1.5 kW × cm⁻², and a full width half-maximum of the point-spread-function of 309 nm.

**Data analysis.** First, super-resolution images were reconstructed and drift-corrected with the render module of Picasso. For the automated evaluation with our MATLAB program, individual structures were selected with Picasso's "pick similar" feature. For the experiments involving the detection of single BS (Figs. 3d and 4), structures were additionally filtered manually by using Picasso's "plot pick" feature. Structures that did not display a correct alignment pattern were discarded. An overview of all selected structures is shown in Supplementary Figs. 9, 10, 12, and 13. Localizations of selected structures were saved as *.hdf5 file and subsequently converted to a *.trace.mat, drift file and mbox file, which then could be imported into the MATLAB program.

The template of each structure was automatically generated after selection of the docking site pattern on the 2D origami map in the template tool. An image was generated by placing Gaussian distributions ($\sigma = 3$ nm, $\sigma = 2$ nm for Fig. 3c or $\sigma = 1$ nm for the single site measurements shown in Figs. 3d and 4) of binding events on the previously defined positions.

For evaluation, a subset with $n$ structures of all selected structures in the *.trace. mat file was used (Fig. 3b, c: $n = 500$; Fig. 3d: $n = 250$; Fig. 4: two datasets were combined and the minimum number of structures that present for each structure type was used, $n = 186$). Next, the software aligned each structure to the structure template: the localizations of each structure were isolated and a super-resolution image of the structure was generated by calculating a 2D histogram of these localizations. The super-resolution image was rotated stepwise in a circle and in each step cross-correlated to the template image. The rotation angle with the highest correlation coefficient was determined. This rotation angle and the corresponding $xy$-shift were then used to transform the localization list in order to align the structure to its template. After alignment, the program counted the number of localizations in a circle with a diameter of 20 nm (Figs. 1c and 3c), 18 nm (Fig. 3b), 6 nm (Fig. 4), or 5 nm (Fig. 3d) around the predefined docking site positions of the template. The number of localizations per docking site for all

structures was displayed in a histogram within the program. This histogram was used to determine the correct detection threshold. This threshold was subsequently used to calculate the presence of each docking site for a given structure. Finally, the program displayed the percentage of detected docking sites for all evaluated structures.

For dual-channel measurements (Fig. 2), 100 structures were manually selected in the reconstructed super-resolution image from Picasso. For each selected structure, the presence or absence of both colors at each docking site was registered.

**Data availability**. All data supporting the findings of this study are available within the paper and its Supplementary Information. All RAW data are available upon request. The MATLAB program and source code for evaluation are available for download at http://www.jungmannlab.org. All sequences of the DNA origami structures are given in Supplementary Tables 6 and 7.

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

## Acknowledgements

We thank Joerg Schnitzbauer, Thomas Schlichthaerle, Peng Yin, and William M. Shih for helpful discussions. We also thank Kimberly A. Cramer for proofreading the manuscript. This work was supported by the DFG through the Emmy Noether Program (DFG JU 2957/1-1), the SFB 1032 (Nanoagents for the spatiotemporal control of molecular and cellular reactions, project A11), the ERC through an ERC Starting Grant (MolMap, Grant agreement number 680241), the Max Planck Society, the Max Planck Foundation, and the Center for Nanoscience (CeNS) to R.J. M.T.S. acknowledges support from the International Max Planck Research School for Molecular and Cellular Life Sciences (IMPRS-LS).

## Author contributions

M.T.S., F.S., and D.H. contributed equally. M.T.S. designed and performed experiments, created software, and wrote the manuscript. F.S. designed and performed experiments and wrote the manuscript. D.H. designed and performed experiments and created software. P.C.N. designed and performed experiments and wrote the manuscript. R.J. conceived of and supervised the study and wrote the manuscript.

## Additional information

**Competing interests:** The authors declare no competing interests.

