## [Peer Review File · Nature Communications]

Reviewers' comments:

Reviewer #1 (Remarks to the Author):

The presented results are interesting and important for the design of DNA origami for precise and efficient binding of nanomaterials and biomolecules. Particularly, it is surprising that there are significant differences at varied locations using short nucleotide sequences and small dye molecules. In addition, it is well-characterized by integrating simulation and the high resolution optical imaging technique. Several important questions have to be addressed before considering the publication:

1. To further verify the effects of strand accessibility and binding efficiency, it would also be interesting to understand the effect the length of docking strands and binding numbers, as it might also affect the downstream attachment of larger guest molecules.
2. It would be clearer to further compare the different detection efficiencies by generating internal edges (i.e., holes) within the 2D origami.
3. Page 4: What is the reason to cause different detection efficiency between 6BS and 3BS at positions 4-6? The staple incorporation efficiency should be ruled out by the 7% offset mentioned in Page 3.
4. Figure 4c and Page 5: It is significant that the heatmap presents the difference in detection efficiency between the central and the edge of the structure. However, there is also uneven distribution of the detection efficiency when comparing different central and edge regions. It would still be worthwhile to confirm the origami-surface interactions did not impose effects to the detection efficiency.
5. Page 5, Line 1-5: It is a little unclear how the accessibility could translate into the incorporation efficiency.
6. Page 5, Line 5, super-subscript reference 26.

Reviewer #2 (Remarks to the Author):

The authors investigate the detectability of DNA handles distributed at different positions within a DNA origami template. The work is well thought out and thorough, and should be very useful to researchers in the DNA origami and DNA paint fields - both as a resource and as an example of how to quantify handle incorporation and accessibility in their own structures. The paper demonstrates an interesting and potentially unexpected observation that detectability is not spatially uniform across their target structure. The authors speculate about a possible cause for this non-uniformity and back this up with simulations which show that the sheet edges are not as stiff as the centre. Although the suggested mechanism is plausible, I believe a lot more work would need to be done to actually prove that this is indeed the reason for the observed non-uniformity. As such the observations in the paper remain fairly phenomenological. Because this is principally a methods paper I do not see this as a major problem.

In technical terms, there is very little to criticise. My biggest concern relates to the ability to separate the different contributions to 'detectability' - namely the incorporation of handles and the subsequent binding of imaging strands to those handles, and whether the ability to distinguish this might be slightly overstated. The authors attempt to quantify this using multiple different imager strands targeted to the same handles to measure the efficiency of imager strand binding and to establish that this is significantly more efficient than handle incorporation. This neglects several mechanisms in which the strand could be incorporated but inaccessible (or less accessible) to all imager strands. One hypothetical mechanism which is particularly pertinent to the observed lower detection efficiencies around the edges of the structures is the possibility that the handle could

become adsorbed onto the substrate and thus unable to hybridize. I think the authors have done enough to establish that detectability is likely to be mostly related to handle incorporation, but not enough to be able to assume this in subsequent discussion.

My concerns could be addressed by focussing on detectability, rather than incorporation, when discussing the following results (particularly the spatial variability - note that they already do this in some places, but not in others), and adding a couple of sentences about the limitations of the detectability=incorporation assumption.

Reviewers' comments:

Reviewer #1 (Remarks to the Author):

The presented results are interesting and important for the design of DNA origami for precise and efficient binding of nanomaterials and biomolecules. Particularly, it is surprising that there are significant differences at varied locations using short nucleotide sequences and small dye molecules. In addition, it is well-characterized by integrating simulation and the high resolution optical imaging technique. Several important questions have to be addressed before considering the publication:

1. To further verify the effects of strand accessibility and binding efficiency, it would also be interesting to understand the effect the length of docking strands and binding numbers, as it might also affect the downstream attachment of larger guest molecules.

Thank you for bringing this up. We have included new data in form of Supplementary Figure 16 where we assayed the effect of docking site length (scaffold length of the newly included Force Clamp origami structure) on strand accessibility/binding efficiency.

Generally, longer docking strands appear to decrease binding probability. As a general rule of thumb, we would suggest to use multiple weak interactions in the form of multiple shorter docking/binding strands to attach molecules of interest, as has already been employed in the literature (see e.g. attachment of gold particles to DNA origami structures [Kuzyk et al, Nature 483, 311-314 (2012) and Gür et al, ACS Nano, 2016, 10 (5), pp 5374–5382]. This furthermore decreases the probability of not being able to bind a guest molecule due to a non-incorporated docking strand.

2. It would be clearer to further compare the different detection efficiencies by generating internal edges (i.e., holes) within the 2D origami.

We thank the reviewer for this useful suggestion. To address this, we have included new experimental data where we directly paint part of the scaffold at the edges of an origami structure in form of the already mentioned newly added Supplementary Figure 16. We refer to our answer to question #4 where we present a more detailed discussion. In short, the new data supports our initial assumption that accessibility of strands is uniform along the structure and even independent of height above the surface.

3. Page 4: What is the reason to cause different detection efficiency between 6BS and 3BS at positions 4-6? The staple incorporation efficiency should be ruled out by the 7% offset mentioned in Page 3.

We thank the reviewer for pointing this out. The error bars shown in Figure 3d were created by taking different sample subsets of origami structures into account. From a statistical point of view, we expect variation depending on the origamis in the subset. Taking this variation into account, the detection efficiency difference is within this deviation.

4. Figure 4c and Page 5: It is significant that the heatmap presents the difference in detection efficiency between the central and the edge of the structure. However, there is also uneven distribution of the detection efficiency when comparing different central and edge regions. It would still be worthwhile to confirm the origami-surface interactions did not impose effects to the detection efficiency.

We thank the reviewer for this suggestion. To study whether origami-surface interaction poses an effect on the detection efficiency we conducted several additional experiments and added the discussion of these experiments to the main text.

First, we measured the detection of a section of the scaffold strand (Supp. Fig 16 a) of the flat rectangular origami. The assumption here is that the scaffold strand should be always present. We chose a section of the scaffold that is anyways single-stranded in the structure design used throughout this study in order to not change the staple pattern. Here we determined that this scaffold loop will be detected in 90% of all cases. To now further investigate if this ought to be an effect of surface interaction, we used a three-dimensional DNA origami structure, the force clamp, (Nickels et al, Science 21 Oct 2016: Vol. 354, Issue 6310, pp. 305-307). This structure positions a section of the scaffold strand between two pillars more than 20 nm above the surface. Here, we mimicked the single-stranded scaffold loop of the flat, rectangular origami and found a 91% detection efficiency, indicating that there is no significant effect arising from surface interactions. To assay where this ~9% "error" stems from, we shortened the scaffold segment (from 49 nt) to 21 nt to mimic an ideally accessible strand (in terms of length, see also our answer to question 1), and achieved a 97% detection efficiency (Supplementary Figure 16c). This measured detection efficiency of 97% suggests that the maximum error arising from accessibility in our measurements is 3%. Finally, we assayed the detection efficiency of a staple hybridized to the scaffold section in the force clamp and found an efficiency of 94%, suggesting that the slight decrease arises from 3% not incorporated staple strands.

5. Page 5, Line 1-5: It is a little unclear how the accessibility could translate into the incorporation efficiency. *We thank the reviewer for bringing this up. We revised our text and added additional information explaining our reasoning here.*

6. Page 5, Line 5, super-subscript reference 26.

We apologize for this mistake. We corrected it accordingly.

Reviewer #2 (Remarks to the Author):

The authors investigate the detectability of DNA handles distributed at different positions within a DNA origami template. The work is well thought out and thorough, and should be very useful to researchers in the DNA origami and DNA paint fields - both as a resource and as an example of how to quantify handle incorporation and accessibility in their own structures. The paper demonstrates an interesting and potentially unexpected observation that detectability is not spatially uniform across their target structure. The authors speculate about a possible cause for this non-uniformity and back this up with simulations which show that the sheet edges are not as stiff as the centre. Although the suggested mechanism is plausible, I believe a lot more work would need to be done to actually prove that this is indeed the reason for the observed non-uniformity. As such the observations in the paper remain fairly phenomenological. Because this is principally a methods paper I do not see this as a major problem.

We thank the reviewer for this positive feedback.

In technical terms, there is very little to criticise. My biggest concern relates to the ability to separate the different contributions to 'detectability' - namely the incorporation of handles and the subsequent binding of imaging strands to those handles, and whether the ability to distinguish this might be slightly overstated. The authors attempt to quantify this using multiple different imager strands targeted to the same handles to measure the efficiency of imager strand binding and to establish that this is significantly more efficient than handle incorporation. This neglects several mechanisms in which the strand could be incorporated but inaccessible (or less accessible) to all imager strands. One hypothetical mechanism which is particularly pertinent to the observed lower detection efficiencies around the edges of the structures is the possibility that the handle could become adsorbed onto the substrate and thus unable to hybridize. I think the authors have done enough to establish that detectability is likely to be mostly related to handle incorporation, but not enough to be able to assume this in subsequent discussion.

My concerns could be addressed by focussing on detectability, rather than incorporation, when discussing the following results (particularly the spatial variability - note that they already do this in some places, but not in others), and adding a couple of sentences about the limitations of the detectability=incorporation assumption.

We thank the reviewer for raising this valuable point about the separation of incorporation and accessibility. To address the questions that arise from this concern, which was also raised by reviewer #1, we performed several additional experiments to further separate incorporation from accessibility. We further added additional text in the manuscript to discuss the limitations of the detectability/incorporation assumption.

REVIEWERS' COMMENTS:

Reviewer #1 (Remarks to the Author):

The authors fully addressed my questions.

Reviewer #2 (Remarks to the Author):

In their revised manuscript, the authors have added an elegant assay of detection vs incorporation through the use of hybridisation of imager strands to a section of the scaffold, eliminating handle incorporation from the equation so that they are only testing accessibility.

I am personally not convinced that the additional experiments fully eliminate all possible causes of variation in accessibility (e.g adsorption of handles onto the substrate), but nonetheless feel that the manuscript provides more than enough information for a reader to come to their own conclusions about this. I believe that the work is ready publication in it's current state.